# Near-Field Coupling Effect Analysis of SMD Inductor Using 3D-EM Model

Gyeong Ryun Choi [1], HyongJoo Kim [1], Yonggi Hong [1], Joosung Hwang [2], Euihyuk Kim [2] and Wansoo Nah [1,*]

1   Department of Electrical and Computer Engineering, SungKyunKwan University, 2066 Seobu-ro, Jangan-gu, Suwon 16419, Republic of Korea; chlrudfbs8@gmail.com (G.R.C.); riding9223@gmail.com (H.K.); hoooh4@gmail.com (Y.H.)
2   LG Electronics, 222 LG-ro, Jinwi-myeon, Pyeongtaek 17709, Republic of Korea; joosung.hwang@lge.com (J.H.); euihyuk.kim@lge.com (E.K.)
*   Correspondence: wsnah@skku.edu; Tel.: +82-31-290-7136

**Abstract:** In this paper, we propose a methodology for analyzing the near-field coupling between two surface mount device (SMD) inductors using a 3-dimensional electromagnetic (3D-EM) model. To develop the 3D-EM model, we first constitute the equivalent circuit of the SMD inductor from the measured impedance and derive the loss tangent using circuit parameters. Secondly, the loss tangent using damped harmonic oscillator model is introduced to extract the effective permeability of core magnetic material in the SMD inductor. The optimization algorithm is used to compare the two loss tangents. Then the effective permeability is used in the magnetic material for the 3D-EM modeling of the SMD inductor. The validity of the proposed 3D-EM model is confirmed by comparing the impedance and S-parameters obtained from both measured and EM-simulated values for the two near-field coupled SMD inductors. Finally, the near-field coupling effects between the two adjacent SMD inductors are visualized in terms of coupling path visualization (CPV) using the proposed 3D-EM model, which demonstrates its usefulness for near-field coupling analysis.

**Keywords:** SMD inductor; loss tangent; effective permeability; near-field coupling

## 1. Introduction

In recent years, there has been an increasing demand for integrated DC-DC converters, leading to efforts to miniaturize components inside the converters. Therefore, surface mount device (SMD) type inductors are replacing conventional through-hole device (THD) type inductors for mounting inside the converter due to their ability to reduce the overall volume and parasitic effects at high frequencies [1].

In high-power environments with large temperature variations, Fe-Si based alloy powder is used for the inductors. According to TDK's power inductor SPM series product overview [2], it is indicated that the product can accept a large current compared to ferrite, has low DC resistance, can be manufactured in a small size, and has superior DC bias characteristics. Moreover, because of their high Curie temperature, these inductors show a small change in characteristics with ambient temperature, have good shielding qualities, and experience minimal magnetic flux leakage due to the coils being integrally molded with metallic magnetic powder, and it is also known that Fe-Si based alloy has a higher magnetic saturation density [3,4]. With all this superiority, Fe-Si based alloy powder has been employed for the magnetic core material of the SMD inductors in the automotive applications with its relatively low permeability (from scores to hundreds) compared to other ferromagnetic materials.

As the distance between components on PCBs decreases to meet the demand for miniaturization of power electronics system, near-field coupling becomes a significant problem. Therefore, several previous studies have been conducted field analysis of inductors using three dimensional electromagnetic (3D-EM) simulation [5,6]. To accurately model

an inductor in a 3D-EM simulation, it is essential to apply magnetic permeability in the core structure of inductor [1]. However, there are cases where inductor manufacturers, due to security reasons, do not provide information about the permeability of the magnetic material. In such situations, engineers are required to extract the permeability of the magnetic material through research and analysis on their own. So far, most of the studies have been performed permeability extraction on the toroid-type inductors [7–13], not on the solenoid-type SMD inductors. For the toroid-type inductors, most of the magnetic flux is confined in the toroid, so that an analytic formula of the inductance is clearly defined, which can be used to extract the complex permeability of the magnetic material [8,9].

In contrast to the existing body of research that predominantly emphasizes toroidal-type inductors, our study is distinct in its specific focus on solenoid-type SMD inductors (referred to as SMD inductors hereafter) for which no analytic formula for inductance exists. Recognizing this gap, we aim to fill it by presenting an innovative and efficient approach that enables the estimation of the effective permeability of the core magnetic material.

Therefore, this paper presents an efficient way to estimate the effective permeability of the core material in the SMD inductors. Firstly, magnetic loss tangent of the SMD inductor is estimated using an equivalent circuit model. Another loss tangent model containing real and imaginary parts of the complex permeability is formulated with the help of the damped harmonic oscillator model [14,15], and then the two loss tangents are compared to find the unknown variables in the damped harmonic oscillator model. The detailed procedure to establish effective permeability will be described. The extracted effective permeability is applied to the magnetic permeability of the SMD inductor in 3D-EM simulation. Additionally, electrical properties such as the permittivity and conductivity of the magnetic material have been set as needed because this information is not provided by the manufacturer to achieve consistency between the actual model and the 3D simulation by tuning the other electrical properties. This comprehensive procedure, which specifically focuses on the extraction of permeability in SMD inductors, addresses an area of research that has remained unexplored until now. By proposing this innovative method, our study offers a significant advancement in the field, allowing for the development of accurate and reliable 3D electromagnetic (3D-EM) models specifically tailored to SMD inductors.

Furthermore, the efficacy of the proposed 3D-EM model is demonstrated through a rigorous validation process. By measuring the S-parameters between two adjacent SMD inductors, we showcase the model's effectiveness in analyzing and understanding the intricate distribution of electromagnetic fields. This capability not only provides valuable insights into the behavior of SMD inductors but also serves as a powerful tool for predicting and addressing potential electromagnetic compatibility (EMC) issues arising from the close proximity of various components during the pre-design stage of product development. Additionally, the proposed model goes beyond mere analysis by enabling the visualization of coupling effects between SMD inductors and neighboring components. This visual representation offers a comprehensive understanding of the complex interactions occurring within the electromagnetic environment, further aiding in the identification and mitigation of EMC-related concerns. The ability to visualize these coupling effects is a crucial feature that enhances the overall utility and practicality of our proposed approach. By extending and refining these aspects in our research, we strive to provide engineers and researchers with an advanced framework for characterizing, simulating, and predicting the electromagnetic behavior of SMD inductors, ultimately contributing to the design and optimization of high-performance electronic systems.

This paper is organized as follows. In Section 2, the magnetic loss tangent based on the equivalent circuit of the SMD inductor is introduced. In Section 3, we propose a method for producing a 3D-EM model, including permeability extraction. We attempt to achieve better consistency between measurement and the 3D-EM model by tuning other electrical properties in Section 4. Finally, in Section 5, we confirm the validity of the produced 3D-EM model and perform near-field coupling analysis using the valid 3D-EM model. We draw conclusions in Section 6.

## 2. Equivalent Circuit Model of SMD Inductor from Impedance Measurement

### 2.1. Impedance Measurement of Wire-Wound Inductor

Before producing a 3D-EM model of the SMD inductor, we measured the impedance of it to evaluate the loss tangent data. As shown in Figure 1a, a 2-port shunt measurement was performed using the SMA connector and lead wire using the vector network analyzer (VNA) E5061B of Agilent. Figure 1b shows equivalent circuit model of SMD inductor, where $Z_1$ and $Z_2$ represent the equivalent impedances of the lead wire and $Z_{DUT}$ for the impedance of the inductor. The sample SMD inductor used in this paper is SPM6530T-2R2M-HZ from TDK [16]. The measured S-parameters can be converted to ABCD parameters using (1), and then the impedance of an inductor can be expressed using (3).

$$\begin{bmatrix} A & B \\ C & D \end{bmatrix} = \frac{1}{2S_{21}} \begin{bmatrix} (1+S_{11})(1-S_{22}) + S_{12}S_{21} & Z_c((1+S_{11})(1+S_{22}) - S_{12}S_{21}) \\ \frac{1}{Z_c}((1-S_{11})(1-S_{22}) - S_{12}S_{21}) & (1-S_{11})(1+S_{22}) + S_{12}S_{21} \end{bmatrix} \quad (1)$$

$$\begin{bmatrix} A & B \\ C & D \end{bmatrix} = \begin{bmatrix} 1 + \frac{Z_1}{Z_{DUT}} & Z_1 + Z_2 + \frac{Z_1 Z_2}{Z_{DUT}} \\ \frac{1}{Z_{DUT}} & 1 + \frac{Z_2}{Z_{DUT}} \end{bmatrix} \quad (2)$$

$$Z_{DUT} = \frac{1}{C} \quad (3)$$

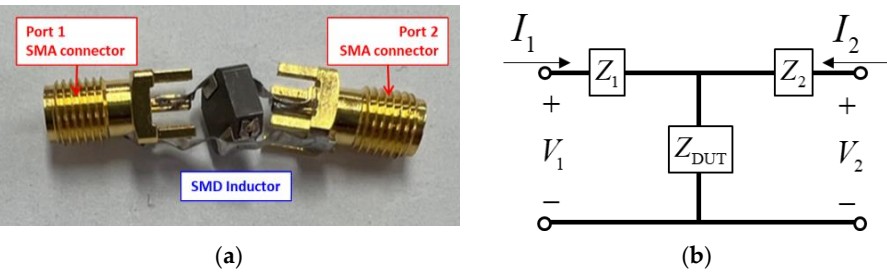

**Figure 1.** (**a**) A picture and (**b**) an equivalent circuit model of 2-port shunt measurement for SMD inductor.

This method has the advantage of accurately measuring the impedance of the device under test (DUT) without the need for de-embedding. Figure 2 compares the measured impedance and the impedance from the data sheet, and one can find that the two impedances coincide very well up to ~100 MHz, which is the highest frequency the manufacturer provides. The self-resonance frequencies (SRF) from the datasheet and the measurement were 39.7 MHz and 40.6 MHz, respectively, corresponding to the error of ~2.5%. The impedance plot shows a typical anti-resonance around 40 MHz: the left side of the self-resonance frequency (SRF) is inductive dominant, and the right side of the SRF is dominated by the stray capacitance due to the coils in the SMD inductor. This leads to an equivalent circuit model in the form of shunt as in Figure 3.

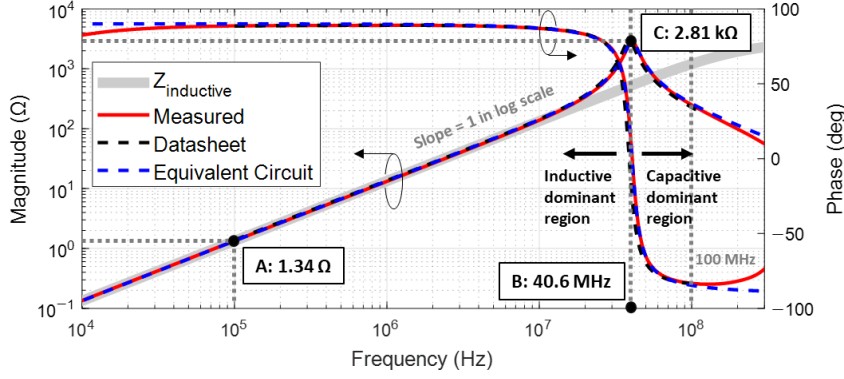

**Figure 2.** Comparison of impedances from measurement and datasheet.

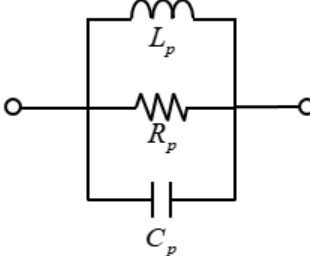

**Figure 3.** Equivalent circuit of SPM6530T-2R2M-HZ.

*2.2. Equivalent Circuit Analysis*

Figure 3 represents an equivalent circuit model of SPM6530T-2R2M-HZ, where $L_p$ represents lossless parallel inductance, $R_p$ for the magnetic loss resistance, and $C_p$ for the stray capacitance of the winding [12]. This shunt equivalent circuit is based on the impedance plot in Figure 2. That is, the impedance from 10 kHz up to 10 MHz turned out to be perfectly linear with slope of 1 in log-scale, which means that the impedance in this region is strongly inductive dominant with constant inductance, and the winding resistance could be negligible. Furthermore, the stray capacitance can be assumed to be constant from the 90 degree of impedance phase. So the values of $L_p$, $R_p$ and $C_p$ can be calculated using the three information in Figure 2: the impedance at A in the inductive dominant region, the anti-resonance frequency at B, and the parallel shunt resistance at C. The formulas are described in (4)–(6).

$$L_p = \frac{1.34\ \Omega}{2\pi \times 100\ \text{kHz}} = 2.13\ \text{uH} \tag{4}$$

$$f_r = \frac{1}{2\pi\sqrt{L_p C_p}}, \ C_p = 7.21\ \text{pF} \tag{5}$$

$$Z_p = R_p \left|\left| j\omega L_p \right|\right| \frac{1}{j\omega C_p} \approx R_p = 2.81\ \text{k}\Omega @ f_r \tag{6}$$

The blue dotted curve is added in Figure 2 to show the impedance from the parallel combination of the calculated $L_p$, $R_p$ and $C_p$ in Figures 2 and 3 shows a good agreement with the measured impedance in both magnitude and phase as well up to ~200 MHz. Note that thick gray line is also added in Figure 2 which corresponds to the impedance of parallel $L_p$ and $R_p$ only as in (7).

$$Z_{inductive} = R_p \left|\left| j\omega L_p \right. \tag{7}$$

**3. 3D-EM Model of SMD Inductor**

*3.1. Inner Structure Acquisition to Produce 3D-EM Model*

Figure 4a shows the structure of the SMD inductor. Single-sided polishing was carried out and the structure of the inductor was identified using microscopic photography, which is shown in Figure 5. The inner coil has 7.5 turns, and the diameter of the coil cross-section is about 400 um on average with a fluctuation of less than 5% from Figure 5a, and the diameter of the inner area of the coil is 3.09 mm from Figure 5b. Peripheral insulator is polyamide and outer thickness of polyamide is 15.47 um. Relative permittivity of the polyamide is set 4.3 with default value in HFSS. In addition, the dimensions of the magnetic material and electrode can be found in the datasheet provided by the manufacturer. Based on these geometrical parameters, a 3D-EM model of the inductor is produced as shown in Figure 4b.

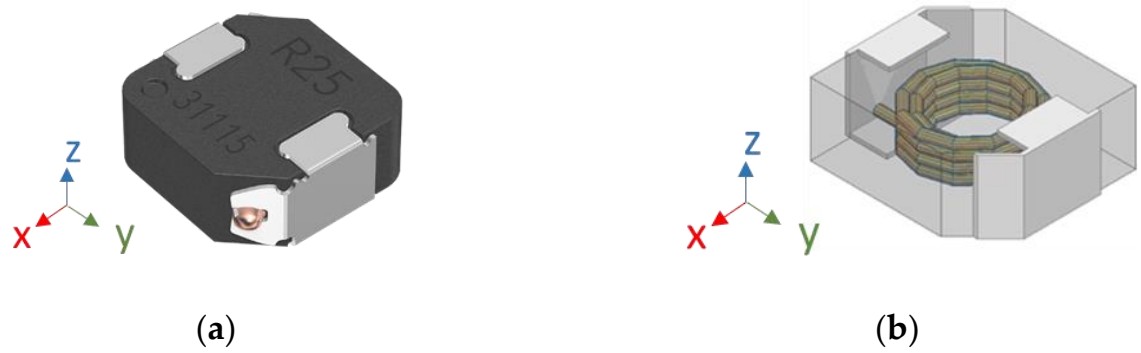

**Figure 4.** (**a**) A picture and (**b**) 3D-EM model of the SMD inductor.

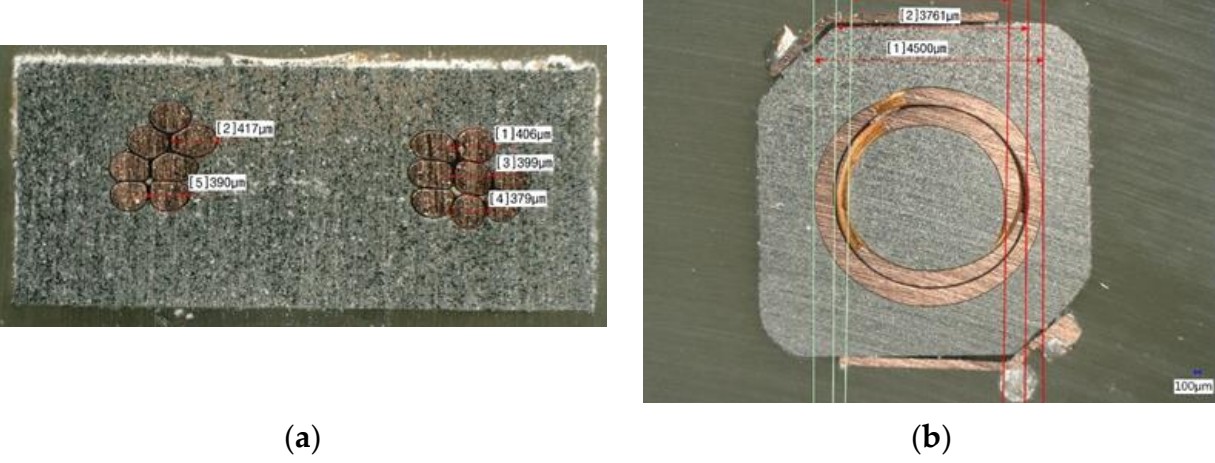

**Figure 5.** (**a**) y-z cross section and (**b**) x-y cross section of SPM6530T-2R2M-HZ.

### 3.2. Permeability Estimation in the Low Frequency Region

Since most manufacturers usually do not provide electrical properties of magnetic materials, we conducted pre-EM simulation by randomly setting the permeability, permittivity, and conductivity of the magnetic material. The purpose of pre-EM simulation is two-folds: one is to check the geometry of SMD inductor in Figure 4b, and the other one is to estimate the permeability in the low frequency range. The estimation of the permeability is possible because the impedance is strongly inductive dominant in the low frequency range as mentioned in Section 2.2, and the impedance can be assumed by $\omega L$ only without any loss.

For the first trial, we set the relative permeability of a magnetic material to be of 100 ($\mu_r' = 100$, $\mu_r'' = 0$), the relative permittivity 12 ($\varepsilon_r = 12$), and the bulk conductivity to 0.01 S/m ($\sigma = 0.01$), which corresponds to '1st 3D-EM simulation'. In Figure 6, The green dotted line shows the impedance obtained from EM simulation, and it shows the impedance of 6.6 Ohm at 100 kHz while the measured impedance is 1.34 Ohm. Noting that inductance is proportional to the permeability of the inductor, we can put the ratio of impedances to be the ratio of inductive permeabilities ($\mu'$), and we get

$$\frac{\omega L_{1st}}{\omega L_{2nd}} = \frac{6.6}{1.34} = \frac{\mu_{r,1st}'}{\mu_{r,2nd}'} \tag{8}$$

$$\mu_{r,2nd}' = 20 \text{ from } \mu_{r,1st}' = 100 \tag{9}$$

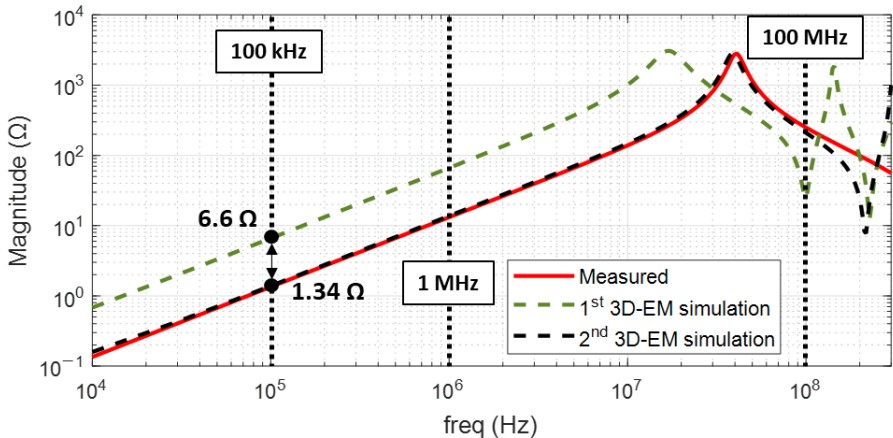

**Figure 6.** Impedances of pre-EM simulations and measurement.

Therefore, for the second trial, the permeability of the magnetic material is revised to 20, and then we obtained black-dotted line in Figure 6. Note that the red curve and the black dotted line coincide very nice especially up to ~100 MHz. This means that the proposed 3D-EM model geometry in Figure 4b with the assumption of $\mu_r' = 20$, $\mu_r'' = 0$, $\varepsilon_r = 12$ and $\sigma = 0.01$ in the magnetic material is valid in the low frequency range, at least up to ~100 MHz, however, we restrict the valid region to be up to 1 MHz, which will be apparent in Figure 7a,b.

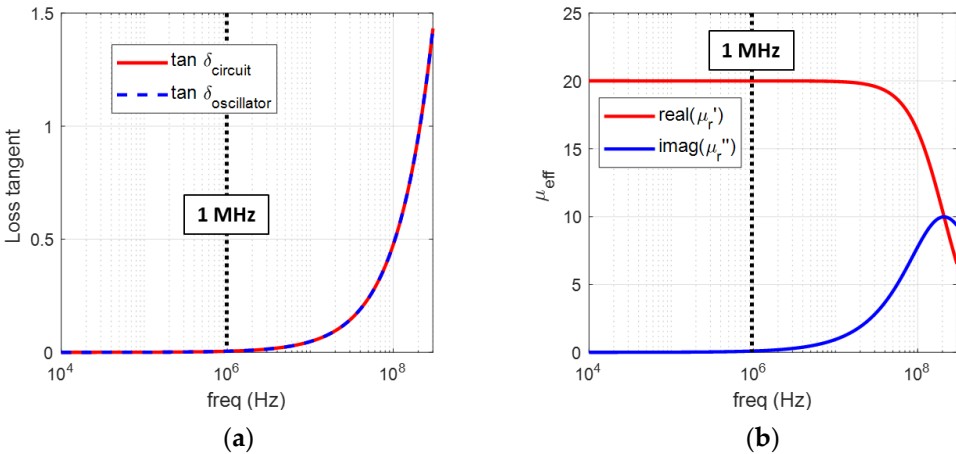

(a)                                                                 (b)

**Figure 7.** (**a**) Comparison of the two loss tangents, and (**b**) Extracted inductive and resistive permeabilities.

### 3.3. The Damped Harmonic Oscillator

The permeability of magnetic material is known to be a frequency-dependent function, with the real and imaginary parts of the permeability having a physical correlation. This relationship is described by the Kramers-Kronig relation, which has been mathematically proven in previous studies [14,17]. In this study, we employ the damped harmonic oscillator model to extract the permeability of the magnetic material, as it is known to satisfy the Kramers-Kronig relation. Damped harmonic oscillator model is originally a formula representing the damping characteristics of a vibrating mass in a fluid. For example, it can be used to calculate the contact force of particles floating on a fluid [15]. By using this model, we are able to effectively represent the frequency response characteristic of magnetic susceptibility as follows [14].

$$\chi_m(\omega) = \frac{1/m}{\omega_r{}^2 - \omega^2 - j\omega\gamma},$$

(10)

where $\chi_m$ is magnetic susceptibility, $m$ is mass, $\gamma$ is damping factor, and $\omega_r = 2\pi f_r$ is angular resonant frequency. Then, the complex permeability is expressed as:

$$
\begin{aligned}
\mu_r(\omega) &= 1 + \chi_m(\omega) \\
&= 1 + \frac{\chi_1}{1 - j(f/k_1) - f^2/k_2{}^2} \\
&= \mu_r' - j\mu_r''
\end{aligned}
\tag{11}
$$

where $\chi_1$, $k_1$ and $k_2$ are unknown variables to be determined, and $\chi_1$ represents the magnetic susceptibility at low frequency region. In principle, the three unknown variables can be determined using another authentic data, and we propose to use $Z_{inductive}$ in (7), which corresponds to the thick gray line in Figure 2 in the frequency range from 10 kHz to 300 MHz. The comparison of the real and imaginary parts in (7) and (11), however, is not possible, because (7) is for inductive impedance and (11) is for the permeability. Still it can be realized that the two loss tangents of (7) and (11) can be compared as needed. This paper proposes a comparison in loss tangent, and (12) is the loss tangent from damped harmonic oscillator model and (13) is the loss tangent from circuit model [18]. It is interesting to note that $\tan\delta_{circuit}$ is $\omega L_p$ divided by $R_p$, not $R_p$ divided by $\omega L_p$ in the circuit model in Figure 3, leading to the increase of $\tan\delta_{circuit}$ as the frequency increases.

$$
\tan\delta_{\text{oscillator}} = \frac{\mu_r''(\omega)}{\mu_r'(\omega)}
\tag{12}
$$

$$
\tan\delta_{\text{circuit}} = \frac{\omega L_p}{R_p}
\tag{13}
$$

The strategy to find permeability of SMD inductor is summarized as follows:

a. The complex permeability formula from damped harmonic oscillator model is set as in (11), and the three unknown variables need to be determined.
b. Another loss tangent from the impedance of circuit model in (13) should be calculated, and the two loss tangents are compared to be equal, for finding the three unknowns using optimization algorithm.

*3.4. Optimization Process*

The goal of the optimization algorithm is to find the optimal combination of $\chi_1, k_1, k_2$ of (11), which makes $\tan\delta_{oscillator}$ go to $\tan\delta_{circuit}$ as close as possible. Several suitable algorithms were investigated, and in this paper, we adopted P2SO algorithm [19–21], the advanced version of the particle swarm optimization (PSO) algorithm [22,23]. The PSO algorithm is efficient in terms of computation and can find optimal convergence values reliably within a short period of time. However, as the PSO algorithm progresses, the diversity of the swarm decreases and the risk of local optima becomes inherent in the algorithm once the rough direction of the solution is determined. To address the issue of premature convergence in PSO, the P2SO algorithm is introduced. P2SO divides the swarm into positive and negative swarms. The positive swarm follows the basic PSO direction to improve the solutions, while the negative swarm moves in the opposite direction of the pbest and gbest positions using a bidirectional repositioning strategy. The positive swarm, similar to the roles of particles in traditional PSO, improves the solutions by iterating generations and following pbest and gbest. The particles in the negative swarm move in the opposite direction, initially following the same direction and velocity as the positive swarm until a certain point (generation), after which they start moving in the opposite direction of pbest and gbest to increase the overall diversity of the swarm. The objective function of the algorithm using P2SO is given by:

$$
Variables : \chi_1, k_1, k_2
$$

$$\textit{Minimize}: |\tan\delta_{\text{circuit}} - \tan\delta_{\text{oscillator}}| < 0.01 \tag{14}$$

$$\textit{Subject to}: \begin{cases} 18 < \chi_1 < 20 \\ 10 \ < k_1, k_2 < 10^{10} \end{cases} \tag{15}$$

The restriction of each variable was set basically by trial and error method, and once the restriction was set the running time was within 60 s using standard workstation (CPU: 2.9 GHz, 6 cores/RAM: 16 GB/GPU: NVIDIA GeForce GTX 1660). The optimized values turned out to be $\chi_1 = 19$, $k_1 = 199.4 \times 10^6$, $k_2 = 906.39 \times 10^6$.

Figure 7a shows $\tan\delta_{circuit}$ and $\tan\delta_{oscillator}$ which was calculated in (11) using the optimized three variables. The two curves coincide very good, which means the optimization process is valid. Also, Figure 7b shows the real and imaginary part of extracted permeability, and they satisfy the Kramers-Kronig relation because damped harmonic oscillator model was used.

However, it should be noted that the permeability we have calculated is not the pure permeability of the core magnetic material in SMD inductor, but includes the permeability of air due to the open structure of the SMD inductor: magnetic flux can reside around the SMD inductor as well as the core magnetic material. That is, the permeability in Figure 7b could be effective permeability rather than permeability of magnetic core material. There are several issues to discuss:

(1)     In Figure 6, where the impedance of the inductor was the parameter to be compared, $\mu'_r = 20$, $\mu''_r = 0$, $\varepsilon_r = 12$ and $\sigma = 0.01$ in the magnetic material seems valid in EM model up to 100 MHz. However, since $\mu''_r$ starts rising from ~1 MHz as in Figure 7, the parameters of $\mu'_r = 20$ and $\mu''_r = 0$ could be valid only up to 1 MHz.

(2)     Loss tangent, $\tan\delta_{circuit}$, came from experiment, which means that $\tan\delta_{circuit}$ includes the loss in the air as well as the loss in the magnetic material. However, noting that loss tangent in the air is null, it is reasonable to say that $\tan\delta_{circuit}$ mainly represents the loss tangent of magnetic material in the SMD inductor. Then, we compared the two loss tangents in (12) and (13) for the estimation of $\mu'_r$ and $\mu''_r$. So, the increase of $\mu''_r$ could be mainly due to the core loss in the magnetic material above 1 MHz in Figure 7b.

(3)     Since $\mu'_r$ and $\mu''_r$ were derived by the comparison of two loss tangents, not by comparison of the $\mu'_r$ and $\mu''_r$ data, it would be instructive to check the magnetic field distribution around the SMD inductor. Figure 8 shows cross sectional view of the magnetic field distribution around SMD inductor at 100, 200 and 300 MHz, respectively, using 3D-EM model. One can see that most of the magnetic fields is restricted inside the core material of the inductor for the three cases.

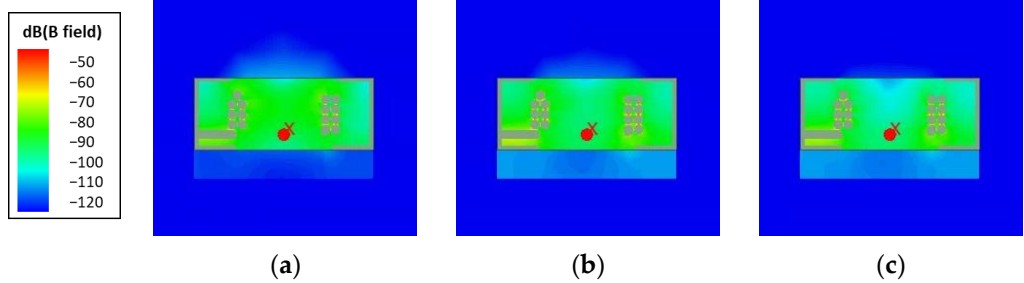

**Figure 8.** Magnetic flux density on y-z cross-section at (**a**) 100 MHz (**b**) 200 MHz (**c**) 300 MHz.

From the above discussions, it could be concluded that $\mu'_r$ and $\mu''_r$ in Figure 7b are effectively estimated complex permeability of the core magnetic material and can be used in the 3D-EM model in Figure 4b.

After applying the effective permeability to the 3D-EM model, it is confirmed that the majority of the magnetic flux density remains within the magnetic material, as depicted in

Figure 8. The distribution of the B-field magnitude beneath the magnetic material stands for the port structure, and can be ignored.

## 4. Tuning of Other Electrical Properties

In addition to magnetic permeability, the permittivity and conductivity of the magnetic material should also be considered in the HFSS simulation. However, information on permittivity and conductivity is not always readily available. Therefore, the authors improved the accuracy of the 3D-EM model by tuning the permittivity and conductivity in the 3D-EM simulation.

### 4.1. Permittivity Tuning

After applying the permeability to the magnetic material, the stray capacitance needed to be adjusted to tune the resonant frequency of the 3D-EM model. The stray capacitance is mainly affected by the space between the windings, and the permittivity of magnetic material and wire insulation as well. Since the structure of the inductor and permittivity of wire insulation were identified in Section 3.1, the permittivity can be adjusted to improve the correlation of the resonant frequency. The effective permittivity of the magnetic material was calculated using the (16).

$$\varepsilon_{EM,2nd} = \varepsilon_{EM,1st} \cdot \left( \frac{f_{r,EM,1st}}{f_{r,m}} \right)^2 = 10.73, \tag{16}$$

where $f_{r,m} = 40.6$ MHz, $f_{r,EM} = 38.4$ MHz are resonant frequencies from measurement and 3D-EM model, and $\epsilon_{EM,1st} = 12, \epsilon_{EM,2nd}$ are the permittivities of the magnetic material in the 3D-EM model before and after modification, respectively.

### 4.2. Conductivity Tuning

To improve the correlation of the impedance magnitude at the resonance frequency, the conductivity of the magnetic material was also adjusted. The conductivity of the magnetic material predominantly influences the value of $R_p$ in the equivalent circuit model (presented in Section 2.2), which limits the impedance magnitude at the resonance frequency. The conductivity is inversely proportional to $R_p$, therefore the conductivity was adjusted to ensure consistency between the simulation and the actual measurement. It was found that by changing the conductivity of the magnetic material from 0.01 to 0.0049 in the 3D-EM model, the impedance magnitude consistency between the simulation result and the measurement was improved. The final EM simulation result is shown in Figure 9, using $\mu$ in Figure 7, $\epsilon_r = 10.73$, $\sigma = 0.0049$.

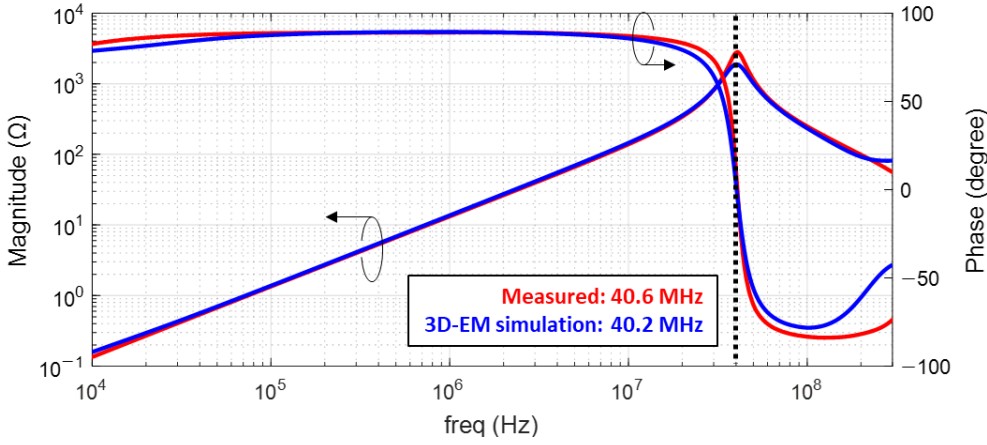

**Figure 9.** Impedance comparison after applying effective permittivity.

## 5. Near Field Coupling Analysis

### 5.1. Validation of SMD Inductor 3D-EM Model

In order to measure the near field coupling between SMD inductors, test board capable of 2 port measurement was fabricated so that adjacent two inductors can be mounted as shown in Figure 10. The distance between two inductors is 1.65 mm.

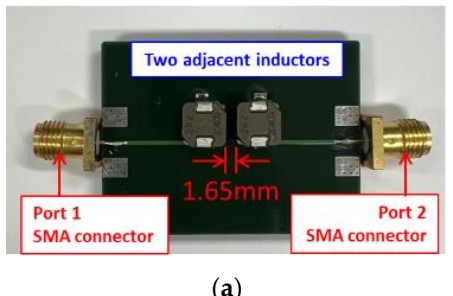
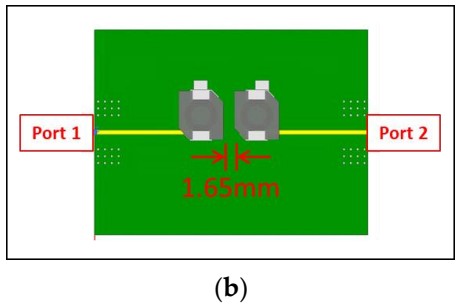

(**a**) (**b**)

**Figure 10.** (**a**) Real and (**b**) 3D-EM model of test board.

The S-parameters of the test board was measured using VNA, and the reflection loss(S11) and the insertion loss(S21) are shown in Figure 11. The agreement between the measurements and 3D-EM simulations is very good up to 300 MHz except some discrepancy observed in S21 above 100 MHz. Figure 11 confirms the validity of the 3D-EM model through a comparison of the S-parameters obtained from the 3D-EM model and the measurements.

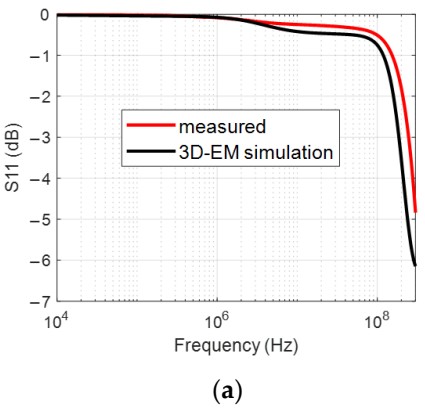
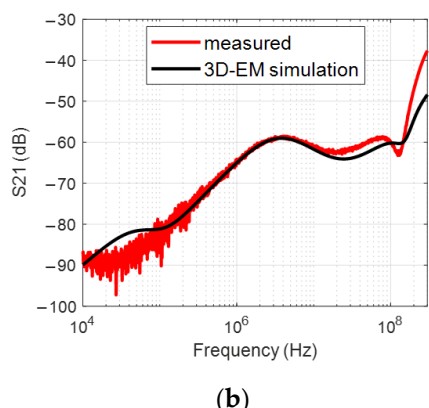

(**a**) (**b**)

**Figure 11.** (**a**) Reflection loss (**b**) Insertion loss.

### 5.2. Coupling Path Visualization

For quantitative analysis of coupling between two inductors, coupling path visualization (CPV) technique [24] was applied. As shown in Figure 12, CPV technique was proposed based on the reciprocity theorem which describes the relationship between fields produced by one source on another and vice versa [25,26]. As shown in Figure 12, the CPV procedure starts with two sub-problems, forward situation and reverse situation. In the forward situation, an emitter is excited, and a receiver is terminated. In the reverse situation, receiver is excited, and the emitter is terminated. Based on the reciprocity theorem, the coupling coefficient density (*CC*) was defined as (17). The superscripts '*fwd*' and '*rev*' represent the forward and reverse problems, respectively.

$$CC = \mathrm{Re}\left\{ \left( \vec{E}^{rev} \times \vec{H}^{fwd} - \vec{E}^{fwd} \times \vec{H}^{rev} \right) \right\} \ (\mathrm{W/m^2}) \tag{17}$$

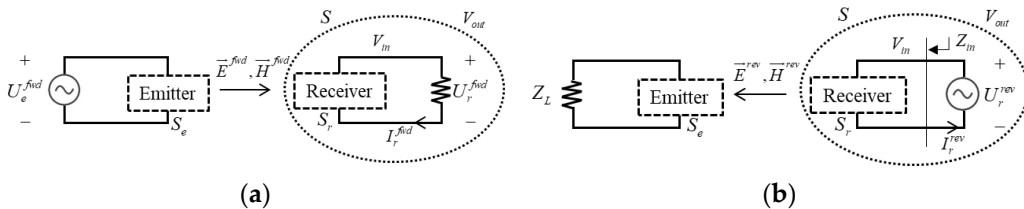

**Figure 12.** Illustration of (**a**) forward situation and (**b**) reverse situation of coupling path visualization techniques.

Visualized *CC* is shown in Figure 13. It is confirmed that *CC* between the two inductors is much greater at 300 MHz than at 100 MHz. In addition, the amount of coupled power on receiver port in forward situation is 95.3 mW at 100 MHz and 143 mW at 300 MHz, which is qualitatively reasonable with S21 of Figure 11b.

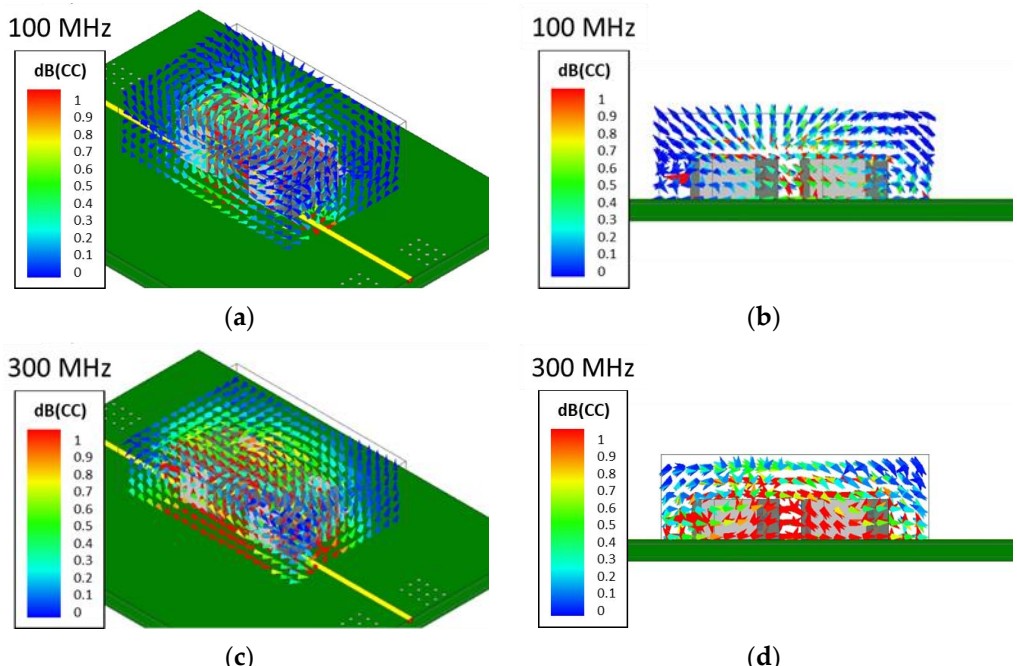

**Figure 13.** CC vector plot (**a**) bird's-eye view (**b**) front view at 100 MHz and (**c**) bird's-eye view (**d**) front view at 300 MHz.

## 6. Conclusions

In this paper, near-field coupling analysis between two adjacent SMD inductors are performed using a 3D-EM simulation, which proved to be valid up to 300 MHz. One of the main proposal and achievements in this study can be the permeability extraction methodology in an SMD inductor, and can be summarized as follows.

(1)　The inductance formula of SMD inductor is not available in an analytic form, so we measured experimental impedance, which was used to develop a circuit model. So the loss tangent of the magnetic material of SMD inductor was effectively defined using the circuit parameters in the developed circuit model.

(2)　The damped harmonic oscillator model, a well-known model that satisfies the Kramers-Kronig relationship, was introduced for comparison with the circuit parameters in terms of loss tangent.

(3)　The two loss tangents, one from the circuit model and the other from damped harmonic oscillator model were compared to extract the complex permeability of SMD magnetic material.

(4) The P2SO optimization method, which was developed based on the PSO algorithm, to avoid a local minimum and the complex permeability was efficiently applied to find the optimized permeability values, which turned out to be valid.

Checking the magnetic field distribution around the SMD inductor numerically, it was confirmed that the obtained complex permeability, which are effective to be precise, could be used for the complex permeability of SMD core material. Applying the obtained permeability to the two adjacent SMD inductors in EM simulation, the coupling effect ($S_{21}$) and impedance were calculated, and compared with the measured ones.

Good agreement was achieved between the measurement and 3D-EM model, as comparing the impedance and near-field coupling between the measurement and EM simulation up to 300 MHz, which demonstrate the validity of the proposed 3D-EM model. Also, we conducted a coupling path visualization to determine the quantity of coupling around the adjacent inductors.

Overall, the proposed 3D-EM model of the SMD inductor can be considered valid and seems to be useful in prior simulation at the product design stage to predict the near-field coupling effect between devices in the future.

**Author Contributions:** Conceptualization, methodology, writing-original draft, G.R.C.; project administration J.H. and E.K.; validation, Y.H. and H.K.; supervision, W.N. All authors have read and agreed to the published version of the manuscript.

**Funding:** This investigation was financially supported by Industry Collaborative Project(S-2021-2828-000) between Sungkyunkwan University and LG Electronics Co., Ltd.

**Data Availability Statement:** Not applicable.

**Conflicts of Interest:** The authors declare no conflict of interest.

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
