# Peer review of "Near-Field Coupling Effect Analysis of SMD Inductor Using 3D-EM Model"

_electronics, doi:10.3390/electronics12132845_

Round 1
Reviewer 1 Report
Authors propose a methodology for analyzing the near-field coupling between two surface mount device (SMD) inductors using a 3-dimensional electromagnetic (3D-EM) model. The results look encouraging and motivating. But there are still some contents, which need be revised in order to meet the requirements of publish. A number of concerns listed as follows:
(1) In the introduction, the authors should clearly indicate the contributions and innovations of this paper.
(2) Please highlight your contributions in introduction.
(3) The computation complexity of the proposed method should be clearly described.
(4) The summary is a little messy, and it is recommended to write it by category.
(5) At Line 104, "the impedance from 10 kHz up to 10 MHz turned out to be perfectly linear with slope of 1 in log-scale." In here, "1" is what?
(6) In section 3.4 of Optimization process, the authors used P2SO algorithm, why? Such as GA, ACO, ABC,... and so on. Please give reason. In addition, how to determine the values of parameters?
(7) he article can be further enhanced by connecting the undergoing work with some existing literatures. For example, https://doi.org/10.1016/j.ins.2023.03.142; http://dx.doi.org/10.1016/j.oceaneng.2022.113424; http://dx.doi.org/10.1145/3513263; http://dx.doi.org/10.1016/j.marstruc.2022.103338 and so on.
(8) What are the limitations behind this study? This topic should be highlighted somewhere in the text of manuscript.
N/A
Reviewer 2 Report
1)This paper investigates the near-field coupling effect analysis of SMD with using 3D-EM model. However, the contents of introduction part are too small and should be riched in detail. The latest literatures should be d in introduction part.
2)The motivations and contributions of this paper should be discussed in detail for near-field coupling effect.
3)The conventional near-field coupling of SMD are not selected as the benchmarks for the purpose of comparison.All simulation parameters have to be justified with proper references in numerical results.
Reviewer 3 Report
This is a well-written paper describing the determination of the effective permeability of surface-mounted inductors to aid in accurate computer simulations. The manuscript describes in detail the steps taken and the various issues they addressed.
I only have two concerns:
First, in the introduction, the authors mention that previous studies analyzed "the radiation characteristics of inductors" and imply that they are doing the same. However, it seems to me that they are really looking at strong inductive coupling between closely spaced components (a low frequency phenomenon, components are within a wavelength of each other), NOT actual EM radiation (a high frequency phenomenon involving EM waves coupling components many wavelengths apart). Later, they state that they are using HFSS, which is a wave based solver. Thus, this part of the introduction is confusing to me and it would be helpful to readers if this issue was clarified.
Second, while the English is very good overall, there are places where it need improvement.
Examples where articles are missing include line 75: "measured THE impedance", line 117: "impedance from THE parallel combination", line 327: "from AN equivalent circuit", line 331: "avoid A local minimum" and so on. In addition, there are singular/plural issues, such as line 81: "measured S-parameterS can be converted to ABCD parameterS using..."
There are also places where sentences are poorly worded, such as line 201, where "makes XXX go to YYY as soon as possible" should be something like "minimizes the difference between XXX and YYY" and line 215 where "curved coincide very good" should be replaced with "curved exhibit good agreement." In line 328, the word "Since" should be dropped.
Overall, this is an informative contribution and I feel it is acceptable after the minor changes suggested above are addressed.
See above
Reviewer 4 Report
1. Describe the reasoning for using P2SO algorithm.
2. Increase the number of frequency points taken for plotting 3D simulation results in Figure 9 and 11. Seems to be very small as compared to measurement results.
Minor changes are required in terms of using grammar. Sometimes use of repeated pronouns in the same sentence makes it difficult to understand and reduces the readability.
Round 2
Reviewer 1 Report
This paper can be accepted now.
Minor editing of English language required
Reviewer 2 Report
The reviewer has no further comments.